# Role of Silica on Clay-Catalyzed Ozonation for Total Mineralization of Bisphenol-A

**DOI:** 10.3390/molecules28093825

**Published:** 2023-04-30

**Authors:** Farida Boudissa, Vasilica-Alisa Arus, Eric-Noel Foka-Wembe, Meriem Zekkari, Rachida Ouargli-Saker, David Dewez, René Roy, Abdelkrim Azzouz

**Affiliations:** 1Nanoqam, Department of Chemistry, University of Quebec at Montreal, Montreal, QC H3C 3P8, Canada; 2Catalysis and Microporous Materials Laboratory, Vasile Alecsandri University of Bacau, 600115 Bacau, Romania; 3Department of Materials Engineering, University of Science and Technology, El M’naouer, B.P. 1505, Bir El Djir, Oran 31000, Algeria; 4École de Technologie Supérieure, Montreal, QC H3C 1K3, Canada

**Keywords:** bisphenol-A, ozonation, mesoporous silica, montmorillonite, hematite

## Abstract

Catalytic ozonation for the total mineralization of bisphenol-A (BPA) from aqueous solution was investigated in the presence of various silica-based catalysts such as mesoporous silica, acid-activated bentonite (HMt) and montmorillonite-rich materials (Mt) ion-exchanged with Na^+^ and Fe^2+^ cations (NaMt and Fe(II)Mt). The effects of the catalyst surface were studied by correlating the hydrophilic character and catalyst dispersion in the aqueous media to the silica content and BPA conversion. To the best of our knowledge, this approach has barely been tackled so far. Acid-activated and iron-free clay catalysts produced complete BPA degradation in short ozonation times. The catalytic activity was found to strongly depend on the hydrophilic character, which, in turn, depends on the Si content. Catalyst interactions with water and BPA appear to promote hydrophobic adsorption in high Si catalysts. These findings are of great importance because they allow tailoring silica-containing catalyst properties for specific features of the waters to be treated.

## 1. Introduction

Wastewaters contain a wide variety of organic pollutants, among which endocrine-disrupting compounds (EDC) were already found to negatively impact human health and biodiversity. These compounds unavoidably oxidize on air-exposed soil–water interfaces into more or less harmful derivatives with the potential spread of toxicity in nature [1,2,3,4,5,6,7,8,9]. Bisphenol-A (BPA), namely 2,2-bis-(4-hydroxyphenyl)propane, was found to display EDC effect on some rodent species [1], affecting animal reproduction and inducing malformations [10,11], aquatic species feminization [12,13,14], some cancers and other diseases [15]. BPA occurrence in waters mainly arises from its use in the plastic packaging of food, beverage, drugs, cosmetics and others [16,17]. BPA concentration reached 21.5 μg/L in urban wastewaters [18], 1.92–11.1 μg·L^−1^ in industrial effluents [19], 0.14–0.98 μg·L^−1^ in treated waters [20] and even 0.5–2 ng·L^−1^ in drinking water [21]. BPA is also still detected in foods, the sera of pregnant women, breast milk, placental tissues and saliva [22,23,24].

The effects of the long-term exposure of food, drinking water, drugs and others to BPA and unavoidable degradation intermediates are still unknown. Thorough BPA mineralization from waters without the residual traces of potentially harmful derivatives has become a major challenge to be addressed. So far, the different approaches tackled in this regard involved more or less conventional methods that often turned out to be quite unsatisfactory due to low removal yields, high energy consumption and use of chemicals [25,26]. Among many oxidation processes [27,28], ozonation is an interesting and convenient route, but ozone alone cannot produce the complete mineralization of the organic pollutants and even less so of its oxidized derivatives. Advanced mineralization requires high ozone and energy consumptions due to low solubility in water. As long as this issue is not addressed, ozonation still cannot be implemented as a viable technique for water treatment [29].

Catalytic ozonation has shown a promising efficiency in EDCs and BPA conversion into supposedly less toxic and more biodegradable products, but some of these turned out to be even more harmful than the parent molecules [2,30,31,32,33,34,35,36,37,38,39,40]. The low efficiency in achieving total pollutant mineralization is often due to the use of inadequately dissolved catalysts. Many catalysts showed effectiveness without necessarily enhancing the production of •OH radicals during ozonation [41], which is rather favored by pH increase [42]. Nevertheless, alkaline polluted wastwaters are not common since most of them rather exhibit acidic to slightly neutral pH. Therefore, investigating the effect of the initial pH induced by a given catalyst in the aqueous media, regarded as the intrinsic pH, is expected to provide valuable information on the catalyst interaction and intrinsic catalytic activity in the ozonation process. The slightest pH adjustment can result in external species that unavoidably influence the ozonation process [43,44]. This can favor an ozonation pathway at the expense of the other (molecular ozone at low pH versus radicals at higher pH) [2,3,34,38] and modify the intrinsic catalyst dispersion or aggregation in aqueous media [45,46,47,48].

Solid catalysts display adsorptive properties that promote surface contact with all ozone species [49], even affording ozone consumptions ca. four times lower than without a catalyst [50]. Dissolved/adsorbed ozone and dispersed gas bubbles can act simultaneously, even affording the total mineralization of refractory derivatives into CO_2_, water and mineral acids [34,38,51]. The lack of knowledge on the interactions of the catalyst surface and organic molecule in aqueous media still remains a major challenge to be tackled. This is why the present work was undertaken.

The effectiveness of a catalyst is favored by large porosity for easy reactant diffusion toward high surfaces with more active sites [52]. So far, activated carbons, zeolites, mesoporous silica and clay minerals have been used in organic pollutant removal [2,34,38,53,54,55,56], but few catalysts have been tested in BPA removal [57,58,59,60,61,62,63,64,65,66,67,68]. The high performances of silica-based catalysts focused interest on acid-activated bentonites and their increased number of silanols, and were explained by the montmorillonite fraction, silica content [46] and exchangeable cations [38,46,69]. Accurate catalyst design requires a deep understanding of clay interactions with the dispersed species based on the key contribution of the silica fraction [70]. This will be examined within a sufficiently wide range of silica contents using various acid-activated bentonites, montmorillonite and Fe^2+^-exchanged forms by comparison to mesoporous SBA15-like silica with large parallel channels [27,71,72,73,74].

Fe^2+^ cation is also expected to have a contribution, mainly acting in the vicinity of the catalyst surface [27,38,46,47]. Given that the natural degradation of biomasses is often related to iron species, some of these species were already tested as catalysts in oxidation processes [27,75]. Since the effect of pH should be essential on iron solubility and clay dispersion, pH evolution during ozonation should govern Fe^2+^–silanol interactions and reversible cation exchange. Thus, the main objective of this work resides in correlating the silica content and catalytic activity with the surface basicity and hydrophilic character. Thermal programmed desorption of CO_2_ and water with zeta potential and particle size measurements allowed the role of silanols to be elucidated in clay catalyst dispersion in aqueous media.

## 2. Results and Discussion

### 2.1. UV-Vis Spectrum Evolution during Non-Catalytic Ozonation

Ozone bubbling induced marked changes in the UV-Vis spectrum of BPA (Figure 1a), as reflected by a marked intensity increase for the 225 nm band (Figure 1b). The latter was attributed to π → π* transitions and suggests the rise of oxidized derivatives such as catechol, acetone, hydroquinone, formaldehyde, acetic, formic, maleic and oxalic acids and related H-bridges. Their persistence even after 40 min ozonation agrees with their high resistance to ozone [67,76]. Simultaneously, the 278 nm band showed a barely detectable intensity decrease for the first 8 min, likely due to the low and slow ozone solubility. This was followed by a pronounced decrease to almost a total depletion after 30 min of ozonation.

This band was assigned to *n → π** transitions involving n electrons of the oxygen atoms of the OH groups on the aromatic ring [24,77] and indicates a pronounced phenolic ring cleavage [78]. Notwithstanding that this indicates a significant decay of the harmful character of BPA, the 278 nm band cannot allow the accurate assessment of BPA depletion during ozonation in spite of the appreciable correlation factor of 0.9998 for the corresponding calibration curve. This is mainly due to possible shading effects of the slight shift toward higher wavelengths and an increasing relative absorbance of the 225 nm band. A much more pronounced shading effect was observed upon clay catalyst addition.

### 2.2. Effect of Clay Catalyst Addition

NaMt addition even induced a UV-Vis spectrum widening beyond 600 nm (Appendix A). This must be due to a shading effect of the formation of slightly tinted polycondensation products, as reported for the reversible production of hypothetical hydroperoxides of ethinylestradiol [79]. Fe(II)Mt addition rather led to an intense 225 nm band, likely due to enhanced hydroxylation (Appendix A). Addition of HMt catalysts induced similar changes in all UV-Vis bands of BPA (Appendix A). The noticeable shift toward higher wavelengths is also expected to shade BPA conversion, thereby explaining the weaker intensity decrease in the 278 nm band for NaMt and Fe(II)Mt beyond 20 min ozonation as compared to non-catalytic ozonation (Figure 2).

This sequence contrasts with that obtained by much more accurate HPLC-UV measurements, inasmuch as the 278 nm showed a much lower intensity decrease as compared to that assessed by HPLC for HMt-1 (Appendix A) and all the catalysts tested (Figure 3). Except for HMt-1, all catalysts displayed a clear and more pronounced decreasing tendency of the HPLC peak area ratio (**dashed line**) compared to the non-catalytic ozonation (**black line**) (Figure 3).

This beneficial effect of the mere presence of solid catalysts was already reported for alumina [33], montmorillonite [27,34,38,46,47,56] and silica [70]. The slight fluctuations of the experimental data as reported to the starting HPLC peak area ratio equal to unity are a common feature of the UV-Vis spectra of almost organic molecules exposed to ozone and is due to enhanced hydroxylation. A wide variety of hydroxylated and carboxylated derivatives are expected to arise not only from BPA but also from its intermediates with potential interactions with each chemical species dispersed in the liquid media, which can affect the UV-detection in HPLC measurements. Such fluctuations completely disappear upon a slight shifts of the detection wavelength from 278 nm to 277.8 or 278.2 or small changes in the retention time.

The new peaks detected at shorter retention times (Appendix A) due to higher polarity derivatives [80] almost completely disappeared after 50–60 min ozonation in the presence of Fe(II)Mt, affording a total mineralization of BPA, as supported by an amount of total organic compounds (TOC) ranging between 1 and 0 ppb. Slightly lower performances with TOC values in the range 2–10 ppb were registered for HMt-4 and HMt-8, which paradoxically produced the highest BPA conversion yields of 86% (HMt-8) and 83% (HMt-4) after only 10 min (Figure 4).

The lower effectiveness of HMt-1, HMt-15, HMt-24 and, to a lesser extent, Fe(II)Mt must not arise from structure collapse given the unaltered crystallinity of all HMt samples [81], but rather from weaker adsorption properties. This result is of great importance because it provides evidence that clay catalysts or at least moderately acid-activated bentonites can produce total mineralization of BPA under optimized conditions to be determined by deeper investigations.

### 2.3. Adsorption on HMt and Hydrophobicity

Here, HMt-1 is assumed to display low specific surface area with less hydrophobic character and then a weaker affinity toward BPA, in agreement with its lowest adsorption yield (6%) (Figure 5). HMt-15, HMt-24 and Fe(II)Mt induced more pronounced pH decrease that promotes clay coagulation flocculation at the expense of the accessible surface (Table 1). Thus, optimum acid attack should result in optimum clay dispersion, demonstrating the key roles of the Si/Al ratio, hydrophobicity, surface acidity and induced pH in the aqueous media. The highest BPA removal yields were registered for HMt-4 and HMt-8 by ozonation (83 and 86%, respectively) (Figure 4) and adsorption (17 and 15%, respectively) (Figure 5).

These performances can be explained not only by their appreciable hydrophilic character reflected by WRC values of 68.2 and 62.3 nmol·g^−1^, respectively, but also by their optimally high pH induced in the aqueous media (4.15 and 4.04, respectively) (Table 1). These two factors are known to promote high clay dispersion in aqueous media and low particle sizes (1.287 and 1.297 μm, respectively), which accounts for high surface accessibility for BPA molecules. Thus, it appears that optimum acid activation leads to optimum clay dispersion, hydrophobicity and active surface accessibility for improved clay–BPA interaction.

### 2.4. Effect of BPA on Clay Dispersion

The reverse proportionality between the particle size and hydrophilic character expressed by the water retention capacity (WRC) as assessed by TPD in the absence of BPA (Figure 6) was in some way expected. Pronounced dealumination and decationization lead to lower surface charge density but a higher amount of hydrophobic Si–O–Si groups that favor clay–clay interaction at the expense of clay dispersion in water. This phenomenon seems to be reversed in the presence of BPA via competitive clay–BPA interaction.

The latter should occur on clay surface, displaying weak affinity toward water expressed in terms of low water retention capacity [46]. BPA molecule should bear no charge at pH levels below 6–7, i.e., lower than its pKa (9.56–10.2) (Appendix A). By analogy to phenol, BPA adsorption on HMt should not involve electrostatic interaction and ion exchange at these pH levels. Hydrophobic interaction of BPA methyl groups is expected to expose the hydroxyl groups to the solution bulk, conferring to clay–BPA particles a hydrophilic character that mitigates clay–clay interaction (Figure 1).

Here, an improvement in clay dispersion was supported by a lower particle size in the presence of BPA as compared to pure water (Table 1). Hydrophobicity inversion induced by BPA adsorption and dispersion enhancement appears to also occur on pure SBA-15 silica as supported by particle size decrease from 1.92 down to 1.58 μm and to a much lesser extent on NaMt (from 0.649 nm to 0.534 μm) (Table 1). The highest WRC of bentonite and Fe(II)Mt (126 and 195 nanomol g^−1^) suggests strong hydrophilic character.

The latter, however, contrasts with the low dispersion of Fe(II)Mt reflected by its highest particle size of 2.45 µm, presumably due to an interlamellar sandwiching effect of bivalent Fe^2+^ cations. This high hydrophilic character should promote BPA adsorption via H-bridges between its hydroxyls and surface silanols, thereby exposing the hydrophobic side (methyl groups) toward the solution bulk as that on silica-rich area of HMt (Figure 1). This is expected to result in a hydrophobic interaction between next-neighboring organoclay lamellae that explains in some way the particle size increase from 1.24 to 1.404 μm for bentonite and from 2.45 to 3.897 μm for Fe(II)Mt.

### 2.5. Role of Silica Content and pH

The initial pH is expected to strongly influence catalyst dispersion in aqueous media [45,46,47,48]. This is mainly due to changes in the behavior of silica or silica moiety regardless of the dispersed silica-containing materials. That is why the present research was focused on the effect of the intrinsic pH induced by catalyst dispersion without pH adjustment to avoid the possible influence of alien anions on the ozonation process [43].

At pH levels higher than the isoelectric point of pure amorphous silica (pH 2) taken as standard [82,83,84], silanols should deprotonate generating negative surface charges. This is well-supported by negative values of the zeta potentials registered for all catalysts, except Fe(II)Mt, due to Fe^2+^ cations shared by pairs of distant exchangeable sites (Table 1). In the meantime, acid activation is also expected to raise the surface density of both silanols (–Si–OH) and siloxy (–Si–O–Si–) groups. Thus, an increasing Si content promotes not only hydrophilic H-bridges, but also hydrophobic clay–clay interactions that reduce the hydrophilic character (Figure 7a) and clay dispersion as supported by the particle size increase (Figure 7b). The higher adsorptive and catalytic activity of HMt samples must be due to a compromise between these two opposite interactions. Thus, moderately acid-activated bentonites must offer minimum particle size for optimum contact surface with BPA within this pH range between 3.86 and 4.27.

This compromise should strongly depend on silica amount and pH. Deeper insights in the role of silica content and its dependence on pH were achieved using pure SBA-15-like silica at previously optimized pH [38,56]. Here, previous pH adjustment was necessary to investigate the behavior of SBA-15 in the vicinity of the point of zero-charge of pure silica (PZC at ca. 2.0–2.3) [70,85,86].

On the one hand, increasing pH from 1.80 up to 3.80 in the presence of 1 g·L^−1^ catalyst induced a slight enhancement of BPA decomposition reflected by a more pronounced decay of the relative intensity (A/Ao) of the 278 nm (Figure 8a). This can be explained by the fact that silanol deprotonation upon pH increase contributes to the ozonation process by enhancing grain dispersion. Prolonged ozonation times beyond 15 min induced a process attenuation, most likely by favoring the rise of acidic products at the expense of phenyl ring cleavage.

On the other hand, raising SBA-15 concentration from 1 to 3 g·L^−1^ SBA-15 at an optimum pH of 2.80 must reduce the surface charge density and repulsive intergranular forces by enhancing silanol protonation [70,85]. This should alter the hydrophobicity inversion induced by BPA adsorption and dispersion enhancement observed at a higher pH (5.27) (Table 1), resulting in particle aggregation that attenuates ozonation. This was reflected by an A/A_o_ increase due to the rise of less oxidized BPA derivatives (Figure 8b). Interestingly, with a complete absence of aluminum, SBA-15 presented an almost similar BPA conversion yield of 90% and beyond, but with a lower catalyst amount (1 g·L^−1^) at pH 2.80 as compared to HMt samples (2 g·L^−1^) at an intrinsic pH of PBA solution. This provides evidence that alumina is not involved in BPA ozonation. However, for a similar catalyst amount (2 g·L^−1^) and initial pH (3.8–3.86), HMt-4 showed slightly higher BPA conversion yields and more pronounced decay than SBA-15 as qualitatively assessed in the relative absorbance of the 278 nm band (Appendix A). This cannot be explained by a lower silica content for HMt samples as compared to SBA-15, but rather by their exfoliation and delamination capacity that improves the diffusion of BPA and ozone molecules toward the solid surface.

### 2.6. Role of Iron

Here, additionally, a previous pH adjustment was necessary to investigate the behavior of hematite above the pH of Fe^3+^ cation precipitation (pH 2). The catalytic activity of Fe^2+^ was supported by LC–ToF-MS measurements, which revealed higher BPA degradation yield of ca. 58.6% and 89.7% after 1 and 5 min, respectively, of Fe(II)Mt-catalyzed ozonation with less intermediate in the retention time range of 2.5–7.0 min as compared to non-catalyzed ozonation (44.8% and 82.8%, respectively (Appendix A)). This result agrees with those already reported for montmorillonite-catalyzed ozonation [2,34,38,40]. Here, the acidic character of Fe(II)Mt reflected by its low CO_2_ retention capacity (CRC) [46] is also supposed to promote BPA adsorption [87]. It The latter must also involve not only cation exchange, but also hydrophobic interaction with the catalyst surface [78]. Cation exchange of Fe(II)Mt with positively charged BPA species must prevail in this pH range. The latter is below the isoelectric point of phenol (6.5) taken as the reference, and should promote free Fe^2+^ cation release. The presence of such cations in the vicinity of the catalyst surface is known to enhance ozonation [40,56].

HPLC measurements already showed that Fe(II)Mt resulted in a higher BPA degradation yield of 24–41% than bentonite (0–25%) after 1–10 min ozonation (Figure 4). This performance accounts for the beneficial role of Fe^2+^ cation, but still remains lower than those produced by all HMt samples under similar conditions (49–86%). A possible explanation should consist of the lower BPA adsorption yield of Fe(II)Mt (14%) compared to HMt-4 (17%) and HMt-8 (15%) (Figure 5). Their higher affinity toward BPA must involve a stronger hydrophobic character as reflected by the their lower WRC and higher adsorption surface as illustrated by finer particle sizes (1.287 and 1.297 μm, respectively) compared to Fe(II)Mt (3.897 μm) (Table 1). This result unequivocally demonstrates that the extent and hydrophobicity of the solid surface have a more significant influence than Fe^2+^ cation.

Deeper insights in the role of iron revealed that BPA adsorbs onto hematite as reflected by decreasing A/A_o_ ratio of BPA at 278 nm during the first 5 min of contact time (Appendix A). This almost similar A/A_o_ decrease for different amounts of hematite (1–3 g·L^−1^) suggests a weak influence of hematite amount. Hematite produced much more pronounced A/A_0_ decrease at 278 nm below 0.5 after 12–13 min ozonation at pH 3.80 as compared to lower values (Appendix A). Qualitative and comparative assessment of the BPA conversion yield based on this A/A_0_ decrease at 278 nm showed that hematite-catalyzed ozonation of BPA allowed for affording a degradation yield of 94% after 25 min ozonation with 1 g·L^−1^ hematite and 89% after only 15 min ozonation with 3 g·L^−1^ hematite (Table 2).

This agrees with the hematite that already showed catalytic activity in the ozonation of other organic molecules [27,88]. This result is of great importance because it demonstrates that low pH values are detrimental most likely due to hematite dissolution into less effective ferric Fe^3+^ cations than its bivalent counterpart [27] and loss in adsorption surface. This research proved that an optimum silica content and pH combined with the presence of Fe^2+^ cations may induce synergy in the catalytic activity of bentonite-based catalysts. These combined factors produced more advanced BPA-removal yields than the individual contribution of each of these factors. This allows for affording not only total BPA degradation as reported by many works [57,58,59,60,61,62,63,64,65,66,67,68], but also thorough mineralization without any trace of its ozonation derivatives [48].

## 3. Materials and Methods

### 3.1. Catalyst Preparation

Bentonite (Sigma-Aldrich, St. Louis, MO, USA) and acid-activated counterparts with 51–140 m^2^·g^−1^ specific surface area (SSA) [81], montmorillonite exchanged with Na^+^ and Fe^2+^ cations (SSA of 54–59 m^2^·g^−1^) [2,34,56,89], SBA-15 silica already synthesized in a previous work and raw natural iron oxide (Hematite, from Chaabet-El-Ballout iron mine, Souk-Ahras (Algeria) [27] were comparatively studied as catalysts. Acid activation of 200 g crud bentonite was achieved in 1000 mL of aqueous 5 M sulfuric acid solution under stirring for 1, 4, 8, 15 and 24 h [81,90,91]. The resulting H^+^-exchanged HMt-1, HMt-4, HMt-8, HMt-15 and HMt-24, respectively, with different Si/Al ratios were repeatedly washed with distilled water until neutral pH was reached and air-dried at 100 °C for 12 h, then smoothly crushed and sifted into 75 μm particle size (200 mesh ASTM).

A montmorillonite-rich material (Mt) was prepared by repeated impregnations of a similar bentonite in 2–5 M aqueous solutions of NaCl (Sigma-Aldrich analytical grade) at 80 °C for 4–5 h under stirring until full ion exchange. The resulting NaMt washed with distilled water and repeatedly dialyzed in cellophane bags was immersed in fresh water until no chloride was detected by AgNO_3_ test, using a ca. 2 wt.% of salt solution which was obtained by dissolving 1.7 g (0.01 mole) of silver salt (Sigma-Aldrich analytical grade) in 98 mL of distilled water. Fe(II)Mt was the result of NaMt impregnation in an aqueous solution of FeCl_2_ 4H_2_O (Merck analytical grade). Both NaMt and Fe(II)Mt, which were selected on the basis of their different behaviors in aqueous media and in ozonation [34,38,46], were further air-dried at 40 °C overnight.

SBA-15 was synthesized by dissolving **6 g** Pluronic P123 copolymer (EO20-PO70-EO20, supplied by Sigma-Aldrich) in **45 g** deionized nanopure water at room temperature, then adding 180 g of 2 M aqueous solution of hydrochloric acid (HCl, 37%, Fluka), stirring for 30 min at 27 °C, and adding 12 g tetraethyl orthosilicate (TEOS, 98%, Sigma-Aldrich) and stirring at 40 °C for 20 h [70,72,74,92]. The resulting white powder was filtered, washed with water, air dried and calcined at 773 K for 6 h at a 1 K·min^−1^ heating rate. Triplicate measurements of the specific surface area presented values in the range 850 m^2^ g^−1^ close to those already reported [72,93].

### 3.2. Material Characterization

All catalysts were already characterized through diverse techniques [27,38,56,72,94]. The latter include powder X-ray diffraction (XRD) (Siemens D5000 equipment, Co-Kα radiation at λ = 1.7890 Å), transmission electron microscopy (TEM) (JEOL JEM-2100F instrument) under a 200 kV accelerating voltage) coupled to energy-dispersion X-ray fluorescence (ED-XRF) and scanning electron microscopy (SEM) (HITACHI S-4300SE/N-VP Emission Scanning Electron Microscope). The specific surface area, pore volume and pore size of the dry materials were evaluated using nitrogen adsorption–desorption isotherms at −195.7 °C recorded on a Quantachrome device equipped with an Autosorb automated gas system control.

Thermal programmed desorption of carbon dioxide (CO_2_-TPD) and water (H_2_O-TPD) allowed the assessment of CO_2_ and water retention capacity (CRC and WRC, respectively) that express the surface basicity and hydrophilic character. TPD measurements were achieved in a tubular oven with a tubular glass reactor containing 40 mg of dry powder of adsorbent/catalyst (0.1–0.25 mm particle size) previously saturated with dry O_2_-free CO_2_ (20–30 mL) under ambient conditions for 60 min without carrier gas. After total evacuation of the non-adsorbed CO_2_ excess at room temperature under a 5 mL·min^−1^ nitrogen stream, as detected at the TPD reactor outlet coupled to a Li-840A CO_2_/H_2_O gas detector (Systech Instruments Ltd., Boca Raton, FL, USA), the heating program was triggered between 20 and 450 °C at a 5 °C·min^−1^ rate [72,95,96].

The initial behavior of the adsorbent/catalyst in aqueous media was evaluated via triplicate measurements of the initial pH, zeta potential and particle size at room temperature (RT) after 30 min contact time in the corresponding aqueous suspension. This was achieved using a ZetaPlus device (BrookHaven Instrument Corp, ZetaPlus/Bl-PALS) and dynamic light scattering (DLS) on a ZetaPlus particle sizer (Brookhaven Instruments, 90 Plus Particles Sizing Software Version 4.20).

### 3.3. Ozonation Tests and Product Analysis

Aqueous 10^−4^ M solution was prepared by dissolving pure and weakly soluble bisphenol-A (≥98%) denoted as BPA supplied by Sigma-Aldrich (Appendix A) in distilled and deionized water under vigorous stirring for 24 h. Samples of 20 mL of this solution were exposed to a 600 mg.h^−1^ ozone bubbling for different times in 50 mL cylindrical tubes in the absence or presence of a 40 mg of catalyst. Ozone was generated by an A_2_Z ozone generator (A_2_Z Ozone Inc., Louisville, KY, USA) from air devoid of CO_2_ and COV, which was obtained through previous purification in columns containing NaOH pills and activated carbon beds. The initial pH was not adjusted uniformly before ozonation in order to avoid favoring: (i) an ozonation pathway at the expense of the other (molecular ozone at low pH versus radicals at higher pH) [2,3,34,38] and (ii) clay catalyst dispersion or aggregation in aqueous media [45,46,47,48]. This is merely because one of the objectives in this work is based on investigating the effect of the intrinsic pH induced in the starting solution in the absence and presence of a catalyst. In addition, pH adjustment would use acids and/or bases whose anions can influence the ozonation process [43]. However, the pH was adjusted before ozonation only for SBA-15 and hematite in order to investigate their behavior around the pH of Fe^3+^ cation precipitation (pH 2) and the point of zero-charge of pure silica (PZC at ca. 2.0–2.3) [70,85,86].

After ozonation and catalyst removal by centrifugation, the supernatant was qualitatively analyzed by UV-visible spectrophotometry only for the qualitative evaluation of the ozonation evolution in time. This was achieved using calibration curves for the main UV-Vis bands of BPA (Appendix A) that allowed for assessing the molar extinction coefficients (Appendix A). This was achieved by means of a Varian-Cary 5000 instrument (Agilent Technologies, Santa Clara, CA, USA) between 190 nm and 800 nm and a 1 cm quartz cell. Additional tests of BPA adsorption involved clay impregnation (40 mg) in 10^−4^ M PBA solution (20 mL of a) for 24 h. Only for a qualitative evaluation of the ozonation progress, a semi-quantitative assessment of BPA depletion in time was achieved using the calibration curve of the 278 nm band with a correlation factor of 0.9998 (Appendix A).

BPA conversion yield by ozonation was quantitatively assessed with a maximum 0.1% standard deviation by high-performance liquid chromatography (HPLC) on a Waters Alliance 2695 system, an Agilent C_18_ column (4.6 mm × 100 mm, 3.5 μm particle size) coupled to a UV detector (Waters 2487, at 278 nm). The mobile phase was an acetonitrile/water mixture (60:40 *v*/*v*) at a 1 mL.min^−1^ flow rate with automatic injection of 20 μL volume. Early ozonation derivatives were identified by liquid chromatography and time-of-flight mass spectrometry (LC-ToF-MS) with an Agilent 1200 Series HPLC device, an Agilent Eclipse Plus C_18_ column (1.8 μm; 3 × 50 mm) and two mobile phases: water–formic acid (A) and acetonitrile–formic acid (B) mixtures [97].

Additional measurements for determining the amount of the total organic carbon (TOC) were achieved by ozonation. For this purpose, the supernatant of the 20 mL ozonized mixture was previously mixed to 1 mL of 0.01 M aqueous HCl solution and then outgassed by N60 grade nitrogen bubbling for 1 h in a sealed flask with an outlet coupled to the Li-840A CO_2_/H_2_O gas detector (Systech Instruments Ltd., USA) used in TPD analysis. After total removal of the CO_2_ absorbed as detected by this device, a 600 mg·h^−1^ ozone bubbling was triggered at 4 h, and the resulting CO_2_ was dried through an on-line acidic clay mineral column (HMt) and then quantitatively assessed on the Li-840A gas analyzer. Ozone was fed by the same A_2_Z generator (A_2_Z Ozone Inc., USA) using air previously stripped of CO_2_ and VOC (volatile organic compounds) through NaOH pills and activated carbon columns. The TOC data were provided within a 0–20,000 ppm range with an accuracy of ≤ 1.5% (sensitivity to water vapor: <0.1 ppm CO_2_//mmol/mol H_2_O).

## 4. Conclusions

The results obtained herein conclude that the silica content contributes to the catalytic activity of montmorillonite in the ozonation of bisphenol-A by offering both adsorption and catalytic sites. Acid activation of bentonite improves the catalytic activity up to a certain level and moderately acid-activated bentonites can produce total mineralization of BPA without traces of persistent derivatives. The fact that Fe(II)Mt gave lower BPA degradation yield than moderately acid-activated bentonite suggests that Fe^2+^ cation is not an essential requirement for advanced BPA degradation. BPA adsorption on HMt samples via hydrophobic interaction appears to be a key step of the global ozonation process. This result unequivocally demonstrates that the extent and hydrophobicity of the solid surface have more significant influence than Fe^2+^ cation. These findings are of great usefulness for a better understanding of the behavior of silica-rich materials such as soils in natural oxidative processes and for designing low cost and natural catalysts for thorough organic contaminant removal from waters.

## Data Availability

Not applicable.

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
