# Peer review of "Role of Silica on Clay-Catalyzed Ozonation for Total Mineralization of Bisphenol-A"

_molecules, 2023, doi:10.3390/molecules28093825_

Round 1

Reviewer 1 Report

This paper investigated ozone oxidation of BPA catalyzed by various silica-based. It was found that the catalytic activity of the silica-based catalysts strongly depended on hydrophobicity. The effect of iron ions on the ozone catalytic process was also studied using Fe2+ ion-exchanged montmorillonite-rich materials and raw natural iron oxide. The study deepened the understanding of silica-based ozone catalysts.

This article could be accepted. However, before that, the following major points should be carefully responded to and revised.

1. There were some very basic grammatical errors in the paper that needed grammatical revision.

2. There were obvious formatting errors in the article, such as the font size in Section 2.2, and the author needs to carefully check the entire text for corrections.

3. The introduction of ozone catalysts was not comprehensive, carbon nanotubes and metal oxides had also been used for ozone catalysis, and relevant references should be added, such as  (Appl. Catal. B: Environ., 2013, 142-143: 533-537), (J. Environ. Chem. Eng., 2022, 10(6): 108726).

4. Information about the reagents and materials used in the experiment should be provided.

5. The pH has a large effect on the active species produced during ozone oxidation (Chemosphere, 2022, 286: 131864) and this should be taken into account as well. In addition, did the authors adjust the pH uniformly before the Fig.3 reaction?

6. In the HMt-24 system in Fig.3, the peak area of BPA showed a significant increase at 30 min, why did this occur?

7. The total organic carbon after the reaction with different catalysts should be tested to obtain a more accurate mineralization degree.

8. Information on the specific surface area of different catalysts should be added to Table.1.

9. The strongly acidic conditions set in the experiment are not common in the actual and experiments under neutral and weakly basic conditions should be supplemented.

Author Response

Reviewer #1:

Reviewer’s general comments

This paper investigated ozone oxidation of BPA catalyzed by various silica based. It was found that the catalytic activity of the silica-based catalysts strongly depended on hydrophobicity. The effect of iron ions on the ozone catalytic process was also studied using Fe2+ ion-exchanged montmorillonite-rich materials and raw natural iron oxide. The study deepened the understanding of silica-based ozone catalysts. This article could be accepted. However, before that, the following major points should be carefully responded to and revised.

Authors’ response

All reviewer’s comments have been considered including the reviewer’s answers to those proposed by the previous two tables. One of these comments referred to the same point as Question in the 2nd table of the reviewer’s report. A point-by-point response to all these comments is herein provided in this revised version of the manuscript.

Question in the 2nd table of the reviewer’s report

Does the introduction provide sufficient background and include all relevant references?

Reviewer’s answer: CAN BE IMPROVED

( )

Authors’ response

Done. Some references have been added and the text of the introduction section has been slightly enriched by new statements. Changes are marked in green color where made.

Reviewer’s specific comments

  1. There were some very basic grammatical errors in the paper that needed grammatical revision.

Authors’ response

Done. The text has been fully spelled and some grammatical revision has been made. The removed or modified grammatical errors and typos are marked in green color where corrected.

Reviewer’s specific comments

  1. There were obvious formatting errors in the article, such as the font size in Section 2.2, and the author needs to carefully check the entire text for correction.

Authors’ response

Done. All sections in this manuscript were already written in Times New Roman, size: 12, except the manuscript general title with a size of 14. Notwithstanding that no exact location and nature of the formatting errors (Font, size) were provided by the reviewer, this comment allowed markedly improving the section referred to.

To comply with this reviewer’s comment, the formatting errors have been checked and corrected according to the author guide throughout the entire manuscript.

Reviewer’s specific comments

  1. The introduction of ozone catalysts was not comprehensive, carbon nanotubes and metal oxides had also been used for ozone catalysis, and relevant references should be added, such as Appl. Catal. B: Environ., 2013, 142-143: 533-537), (J. Environ. Chem. Eng., 2022, 10(6): 108726).

Authors’ response

Done. The comment refers to the same point as Question in the 2nd table of the reviewer’s report. Thus, the same answer is provided below:

The suggested references and others have been added and the text of the introduction section has been slightly enriched by new statements. Changes are marked in green color where made.

Reviewer’s specific comments

  1. Information about the reagents and materials used in the experiment should be provided.

Authors’ response

Done. Additional details with reagent Brands and purity grade have been provided in the experimental section. The added details are marked in green color.

Reviewer’s specific comments

  1. The pH has a large effect on the active species produced during ozone oxidation (Chemosphere, 2022, 286: 131864) and this should be taken into account as well. In addition, did the authors adjust the pH uniformly before the Fig.3 reaction?

Authors’ response

Yes indeed, the pH is a key-factor that determines not only the ozonation pathway and predominant oxidizing species (molecular ozone at low pH versus radicals at higher pH) but also the clay catalyst dispersion and behavior in aqueous media. The effect of pH was already discussed for correlating the data of Figure 5 for adsorption and Table 1 for clay dispersion in section 2.3 (Former 3.3) of this manuscript.

The pH was adjusted before ozonation only for SBA-15 and hematite in order to investigate their behavior around the point of zero-charge of pure silica (PZC at ca. 2.0-2.3) and pH of Fe3+ cation precipitation (pH 2). The pH was not uniformly adjusted merely because:

  1. the effect of the intrinsic pH induced in the starting solution in the absence and presence of catalyst was targeted as a specific objective in this work.
  2. pH adjustment would require the use of acids and/or bases, which unavoidably introduce anions with potential effects on the ozonation process, as supported by the very reference suggested by the reviewer (Influence of anions on ozonation of bisphenol AF: Kinetics, reaction pathways, and toxicity assessment, Chemosphere, 2022, 286: 131864).

However, to comply with this reviewer’s comment and for the sake of clarity, some of the already provided statements have been rephrased and enriched by additional references in this section and in sections 2.4 and 2.5. The suggested references have been added. See changes marked in green color in the 1st and 2nd paragraphs of the introduction section (Pages 2-3) and in the 1st paragraph of sections 2.5. Role of silica content and pH (Page 14) and 3.3. Ozonation tests and product analysis (Page 21).

Reviewer’s specific comments

  1. In the HMt-24 system in Fig.3, the peak area of BPA showed a significant increase at 30 min; why did this occur?

Authors’ response

This apparent increase is not so ‘’significant’’, being a mere and slight fluctuation of the experimental data as reported to the starting HPLC peak area ratio equal to unity. Such fluctuations of the experimental data is a common feature of the UV-Vis spectra of almost organic molecules exposed to ozone and is due to enhanced hydroxylation. A wide variety of hydroxylated and carboxylated derivatives are expected to arise not only from BPA but also from its intermediates with potential interactions between each other can affect the UV-detection in HPLC measurements. Such a detection mode is not quite accurate for a relatively broad peak at an average retention time after prolonged ozonation time of 20 min and beyond. Such a fluctuation totally disappears after additional measurements at slight shifted detection wavelength from 278 nm to 277.8 or 278.2 or of the retention time. See the improved version of this figure.

To comply with this reviewer’s comment and for the sake of clarity, some of these statements have been incorporated as a 2nd paragraph in section 2.2. Effect of clay catalyst addition.

Reviewer’s specific comments

  1. The total organic carbon after the reaction with different catalysts should be tested to obtain a more accurate mineralization degree.

Authors’ response

Done. Conventional TOC measurement usually provides data accuracy barely reaching values below 2%, since 1% accuracy was only reported for UV oxidation combined to differential conductivity or potentiometry. Unless special care is previously taken, conductivity detection is known to be a major source of error given the possible interference of the electrical properties of the acids generated by ozonation. To avoid this major shortcoming, Our TOC measurements were achieved by ozonation for 5 hours using the A2Z generator (A2Z Ozone Inc., USA) and CO2 quantification by means of the Li-840A CO2/H2O Gas detector (Systech Instruments Ltd., USA) used in TPD analysis. The TOC data were provided within a 0-20.000 ppm range with an accuracy ≤ 1.5 %.  

However, to comply with this reviewer’s comment, and notwithstanding that we strongly believe that TOC is by far less accurate (1-3 %) than HPLC measurements (max 0.5 %), we provide herein additional information related to TOC as requested by the reviewer.

Reviewer’s specific comments

  1. Information on the specific surface area of different catalysts should be added to Table.1.

Authors’ response

Unless operating in dry media, except for SBA-15 which display rigid to slightly flexible framework with almost constant porosity, the concept of specific surface area (SSA) is not relevant when dealing with aqueous suspension in aqueous media. The SSA significantly varies from its starting value in dry media to various values in aqueous media depending on the dispersion grade and subsequently on the particle size and grain aggregation, In aqueous media, the particle size is governed by clay dispersion aggregation equilibrium which strongly depend on the clay concentration, presence of other chemical species and pH. That is why the concept of SSA for clay materials is replaced by that of particle size which provides a clearer and more accurate assessment of clay dispersion in aqueous media. Some of these statements were already provided in section 2.5 (former section 3.5).

However, to comply with this reviewer’s comment, the specific surface area (SSA) of SBA-15 (850-900 m2 g−1), of bentonite and acid-activated counterparts in dry state (40 up to almost 140 m2 g-1) and of dry NaMt (54 m2 g-1) have been added in table 1 and related section. For the sake of clarity, some of the already provided statements have been slightly rephrased. See changes marked in green color in section 2.3, 2.4 and 2.5.

Reviewer’s specific comments

  1. The strongly acidic conditions set in the experiment are not common in the actual and experiments under neutral and weakly basic conditions should be supplemented.

Authors’ response

Ozonation under neutral and weakly basic conditions was never targeted by our 20 years of research and even less by the present work. Why? Merely because wastewaters are mostly slightly alcaline to acidic and also because of the rise of undesired Fe3+ cations which should unavoidably influence the Fe3+ 1  Fe2+ equilibrium and consequently the catalytic activity of targeted Fe2+ cations.

Our work focuses on the intrinsic pH of BPA solutions alone (Intrinsic pH of 10-4 BPA solution: 5.64. and in the presence of NaMt and Fe(II)Mt aqueous suspensions (pH 8.77-2.38). In other words, the investigated pH range include not only the pKa’s of both in-plane (8.5) and out-of-plane silanols (5.6) but also the

isoelectric point of BPA that is assumed to be close to similar as that of phenol (pH 6.4). In this pH range, BPA adsorption, which is key step in the global ozonation process, depends on the critical coagulation concentration, and is expected to involve at least three types of interactions, i.e., hydrophobic, H-bridges and ion-exchange. BPA-silanol interaction assessment was the major objective of the present work.

However, to comply with this reviewer’s comment, some of these statements have been added in section 2.3. in order to justify the pH range investigated.

Reviewer 2 Report

The manuscript presents an important amount of experimental data but it is shown in a way that, in my opinion, is not clear. Improvement of the wording and structure of the text is highly recommended before being accepted.

I have some questions, but in general for me it was difficult to follow the text:

·         In the introduction section it is mentioned the importance of BPA mineralization but total organic carbon measurements are not reported. The majority of the analysis is based on UV-VIS spectra, and I am not sure of how precise this information is to interpret the slight differences between the behavior of different catalysts.

·         The quality of the figures is bad.

·         I think surface area and porosity should be reported and connected to the results obtained.

·         Page 8 line 233. It is mentioned that HMt-4 and HMt-8 present the highest WRC but they show intermediate values, higher than HMt-15 and HMt-24 but lower than HMt-1, Fe(II)Mt, NaMt and bentonite.

·         Page 9 line 261. First you are referring to SBA-15 and NaMt and then in line 264 by “their” you are referring to bentonite and FeMt. This is confusing.

·         I think it would be valuable to add a section where you explain in detail the catalytic mechanism for each material under the pH condition chosen. You did not mention the presence of hydroxyl radicals. How did you discard this possibility?

·         Figure 8, b) You mentioned a clay catalyst but in the text you are referring to SBA-15.

·         Under all the conditions studied the blank tests without catalyst under the same pH conditions should be presented.

Author Response

Reviewer #2:

Reviewer’s general comments

The manuscript presents an important amount of experimental data, but it is shown in a way that, in my opinion, is not clear. Improvement of the wording and structure of the text is highly recommended before being accepted. I have some questions, but in general for me it was difficult to follow the text:

Authors’ response

Done. Changes in the paper structure have been made in order to better understand the main specific objectives targeted by the present research. Each subsection title is a targeted specific objective.

For the sake of clarity, some statements have been modified in different locations throughout the manuscript and according to each reviewer’s comments. These changes are marked in green color.

Question in the first table of the reviewer’s report

Extensive editing of English language and style required.

Authors’ response

Done. A full spelling of the text has been performed and some style issues have been addressed to improve the English standard. Changes are marked in green color where made throughout the whole revised manuscript.

Question in the 2nd table of the reviewer’s report

Are the methods adequately described?  Reviewer’s answer: CAN BE IMPROVED

( )

Authors’ response

Done. Improvements have been brought in this regard by incorporated new details (marked in green color).

Reviewer’s specific comments

 In the introduction section it is mentioned the importance of BPA mineralization, but total organic carbon measurements are not reported.

Authors’ response

Done.

Done. Conventional TOC measurement usually provides data accuracy barely reaching values below 2%, since 1% accuracy was only reported for UV oxidation combined to differential conductivity or potentiometry. Unless special care is previously taken, conductivity detection is known to be a major source of error given the possible interference of the electrical properties of the acids generated by ozonation. To avoid this major shortcoming, Our TOC measurements were achieved by ozonation for 5 hours using the A2Z generator (A2Z Ozone Inc., USA) and CO2 quantification by means of the Li-840A CO2/H2O Gas detector (Systech Instruments Ltd., USA) used in TPD analysis. The TOC data were provided within a 0-20.000 ppm range with an accuracy ≤ 1.5 %.  

However, to comply with this reviewer’s comment, and notwithstanding that we strongly believe that TOC is by far less accurate (1-3 %) than HPLC measurements (max 0.5 %), we provide herein additional information related to TOC as requested by the reviewer.

Reviewer’s specific comments

The majority of the analysis is based on UV-VIS spectra, and I am not sure of how precise this information is to interpret the slight differences between the behavior of different catalysts.

Authors’ response

UV-visible spectrophotometry was only used for qualitative evaluation of the ozonation evolution in time. This statement was already provided in the manuscript in the paragraph after Figure 1. Yes, indeed slight evolution of the shape and intensity of the main UV-Vis band are quite useful for distinguishing the product distribution produced by ozonation with two different catalysts. This provided sufficiently rich information to state, for instance that hydroxylation is more intensive on one catalyst than on another. Only for qualitative evaluation of the ozonation progress, a semi-quantitative assessment of BPA depletion in time was achieved using the calibration curve of the 278 nm band with a correlation factor of 0.9998. Only four among the eight figures of this manuscript were based on UV-Vis measurements. As a reminder of what was already and repeatedly stated in the manuscript,

BPA conversion yield was quantitatively and accurately determined with a max 0.5 % accuracy on the basis of measurements through high-performance liquid chromatography (HPLC).

Reviewer’s specific comments

The quality of the figures is bad.

Authors’ response

Done. All figures have been improved and the legends and axes font and size have been re-sized uniformly as requested.

Reviewer’s specific comments

I think surface area and porosity should be reported and connected to the results obtained.

Authors’ response

Done. Since this comment is similar to that of reviewer 1, the same answer is provided below.

Unless operating in dry media, except for SBA-15 which display rigid to slightly flexible framework with almost constant porosity, the concept of specific surface area (SSA) is not relevant when dealing with aqueous suspension in aqueous media. The SSA significantly varies from its starting value in dry media to various values in aqueous media depending on the dispersion grade and subsequently on the particle size and grain aggregation, In aqueous media, the particle size is governed by clay dispersion aggregation equilibrium which strongly depend on the clay concentration, presence of other chemical species and pH. That is why the concept of SSA for clay materials is replaced by that of particle size which provides a clearer and more accurate assessment of clay dispersion in aqueous media. Some of these statements were already provided in section 2.5 (former section 3.5).

However, to comply with this reviewer’s comment, the specific surface area (SSA) of SBA-15 (850-900 m2 g−1), of bentonite and acid-activated counterparts in dry state (40 up to almost 140 m2 g-1) and of dry NaMt (54 m2 g-1) have been added in table 1 and related section. For the sake of clarity, some of the already provided statements have been slightly rephrased. See changes marked in green color in section 2.3, 2.4 and 2.5.

Reviewer’s specific comments

Page 8 line 233. It is mentioned that HMt-4 and HMt-8 present the highest WRC but they show intermediate values, higher than HMt-15 and HMt-24 but lower than HMt-1, Fe(II)Mt, NaMt and bentonite.

Authors’ response

Done. Yes, indeed. There was confusion. The sentences referred to by the reviewer have been identified and rephrased to avoid confusion as follows:

‘’These performances can be explained not only by their appreciable hydrophilic character reflected by WRC values of 68.2 and 62.3 nmol.g-1, respectively but also by their optimally high pH induced in the aqueous media (4.15 and 4.04, resp.)’’. See change in the 1st sentences of the 1st paragraph after Table 1.

Reviewer’s specific comments

Page 9 line 261. First you are referring to SBA-15 and NaMt and then in line 264 by “their” you are referring to bentonite and FeMt. This is confusing.

Authors’ response

Done.

Yes, indeed. This mistake, due to an inadvertently unintentionally cut sentence, has been corrected in the last paragraph of section 2.4. The entire last paragraph has been rewritten in page 13.

Reviewer’s specific comments

I think it would be valuable to add a section where you explain in detail the catalytic mechanism for each material under the pH condition chosen. You did not mention the presence of hydroxyl radicals. How did you discard this possibility?

Authors’ response

Done.

At the acidic pH induced by most catalysts, except for bentonite (slightly alkaline pH), the ozonation process should preponderantly involve the action of molecular ozone. Radicalic mechanisms should prevail in alkaline pH. In other words, effective ozonation does not necessarily involve radicals, as illustrated by ample literature and was not targeted by the present research. The effect of pH was already discussed for correlating the data of Figure 5 for adsorption and Table 1 for clay dispersion in section 2.3 (Former 3.3) of this manuscript. The pH was adjusted before ozonation only for SBA-15 and hematite in order to investigate their behavior around the point of zero-charge of pure silica (PZC at ca. 2.0-2.3) and pH of Fe3+ cation precipitation (pH 2). The pH was not uniformly adjusted merely because:

  1. the effect of the intrinsic pH induced in the starting solution in the absence and presence of catalyst was targeted as a specific objective in this work.
  2. pH adjustment would require the use of acids and/or bases, which unavoidably introduce anions with potential effects on the ozonation process, as supported by the very reference suggested by the reviewer (Influence of anions on ozonation of bisphenol AF: Kinetics, reaction pathways, and toxicity assessment, Chemosphere, 2022, 286: 131864).

However, to comply with this reviewer’s comment and for the sake of clarity, and notwithstanding that specific discussion of the catalytic interaction according to the pH level was already provided, some of these have been reformulated with the incorporation of new references for a better understanding. See changes marked in green color in the 1st and 2nd paragraphs of the introduction section (Pages 2-3) and in the 1st paragraph of sections 2.5. Role of silica content and pH (Page 14) and 3.3. Ozonation tests and product analysis (Page 21).

Reviewer’s specific comments

Figure 8, b) You mentioned a clay catalyst but in the text, you are referring to SBA-15.

Authors’ response

Done.

Yes, indeed. This mistake, due to an inadvertently unintentionally cut sentence, has been corrected. The caption of Figure 8 has been rewritten. See changes marked in green color in page 15.

Reviewer’s specific comments

Under all the conditions studied the blank tests without catalyst under the same pH conditions should be presented.

Authors’ response

All experiments were already compared with blanks achieved in the absence of catalysts, but not necessarily at the same pH in order to maintain the intrinsic pH of BPA solution before and after catalyst addition. These pH values should be different and are a key-factor of the intrinsic reactivity of ozone and catalytic activity of the investigated materials. This was our starting approach for achieving this research. pH adjustment is another approach targeting other objectives.

In the present work, pH adjustment was avoided for the reasons explained above as our answer to the reviewer’s comment. Here is below an extract from this answer:

The pH was not uniformly adjusted merely because:

  1. the effect of the intrinsic pH induced in the starting solution in the absence and presence of catalyst was targeted as a specific objective in this work.
  2. pH adjustment would require the use of acids and/or bases, which unavoidably introduce anions with potential effects on the ozonation process, as supported by the very reference suggested by the reviewer (Influence of anions on ozonation of bisphenol AF: Kinetics, reaction pathways, and toxicity assessment, Chemosphere, 2022, 286: 131864).